# COVID-19 Public Health Measures and Patient and Public Involvement in Health and Social Care Research: An Umbrella Review

**DOI:** 10.3390/ijerph20064887

**Published:** 2023-03-10

**Authors:** Negin Fouladi, Nedelina Tchangalova, Damilola Ajayi, Elizabeth Millwee, Corinne Lovett, Alana Del Sordi, Samantha Liggett, Malki De Silva, Laura Bonilla, Angel Nkwonta, Leah Ramnarine, Allyssa Munoz, Kate Frazer, Thilo Kroll

**Affiliations:** 1Department of Health Policy and Management, School of Public Health, University of Maryland, College Park, MD 20742, USA; 2STEM Library, University of Maryland, College Park, MD 20742, USA; 3School of Nursing, Midwifery and Health Systems, University College Dublin, D04 V1W8 Dublin, Ireland

**Keywords:** SARS-CoV-2, COVID-19, communicable diseases, pandemic, disease transmission, public health measures, patient and public involvement and engagement

## Abstract

An umbrella review of previously published systematic reviews was conducted to determine the nature and extent of the patient and public involvement (PPI) in COVID-19 health and social care research and identify how PPI has been used to develop public health measures (PHM). In recent years, there has been a growing emphasis on PPI in research as it offers alternative perspectives and insight into the needs of healthcare users to improve the quality and relevance of research. In January 2022, nine databases were searched from 2020–2022, and records were filtered to identify peer-reviewed articles published in English. From a total of 1437 unique records, 54 full-text articles were initially evaluated, and six articles met the inclusion criteria. The included studies suggest that PHM should be attuned to communities within a sociocultural context. Based on the evidence included, it is evident that PPI in COVID-19-related research is varied. The existing evidence includes written feedback, conversations with stakeholders, and working groups/task forces. An inconsistent evidence base exists in the application and use of PPI in PHM. Successful mitigation efforts must be community specific while making PPI an integral component of shared decision-making.

## 1. Introduction

Over the past decade, there has been an increased emphasis on the importance of patient and public involvement (PPI) in health and social research as it provides alternative views and insights into the needs of healthcare users to improve the quality and relevance of research [1]. PPI integration into health and social care research gained momentum before the COVID-19 pandemic. However, the rapid response to the pandemic resulted in PPI being viewed as non-essential, leading to its minimal inclusion in research and, thereby, minimizing the contribution of patients, the public, and, particularly, minority groups in helping find solutions to the pandemic crisis [1].

The concept of patient and public involvement refers to conducting research ‘with’ or ‘by’ the public rather than ‘to’, ‘about,’ or ‘for’ them. Patients, potential patients, caregivers, and people who use health and social care services, as well as the representatives of organizations representing such people, are considered the public [2]. PPI creates an active partnership with patients, healthcare providers, members of the public, and researchers, with the goal of influencing and shaping research in a manner that is relevant and inclusive to individuals of various backgrounds and ethnicities [3]. The integration of PPI has been demonstrated to produce positive impacts on the quality and appropriateness of research during all stages of the research process, including the development of user-focused research objectives and questions, user-friendly information, improvements in recruitment strategies, the consumer-focused interpretation of data, and the enhanced implementation and dissemination of study results [4]. The National Institute for Health and Care Research (NIHR) has been instrumental in firmly entrenching PPI inclusion across publicly funded research in the United Kingdom, and a similar prioritization of PPI has been observed in other parts of Europe, Canada, the United States, and Australia [5]. Ireland, too, has adopted a nationwide approach to embedding PPI in all academic health and social care research through the creation of a network on PPI [6].

Patient and public involvement (PPI) is not a recent phenomenon. It shares attributes with participatory research, community and citizen engagement, consumer involvement, and community empowerment. While we are citing the NIHR definition, many other conceptualizations exist. What they have in common is the central, co-producing, and participatory role of patients, as well as members of the public and communities, in the research process.

Although the past decade has shown the great expansion of the PPI evidence base, the reporting of PPI and its impact is inconsistent as well as lacking in information about the context and process of patient and public engagement. The inadequate reporting of PPI creates significant barriers to advancing the adoption of PPI strategies in health and social care research as it produces substantial challenges in the appraisal, interpretation, and synthesis of evidence for systematic reviews, as well as the ethical implications of reporting PPI to improve quality and transparency for the use of research findings [5]. Particularly during the pandemic, a high number of systematic reviews relating to COVID-19 were conducted with PROSPERO records indicating findings that were not replicable and poorly reported [7].

In order to implement strategies to address the COVID-19 pandemic, such as patient and public involvement and the utilization of public health measures, it is critically important to have consistent and reproducible findings. On the societal level, PPI-generated trust and acceptance in research, greater benefits for the community, new and improved services, valuable changes in practice/partnership lead to positive changes and outcomes, and finally, improved relationships between professionals and communities [8].

Public health measures (PHM) aim to reduce the transmission, severity of illness, and death and are critical strategies to address pre-COVID-19 pandemic outbreaks, such as severe acute respiratory syndrome (SARS) and influenza. PHM strategies prior to the COVID-19 pandemic included personal and pharmaceutical measures such as travel, social distancing, the use of personal protective equipment, and medications [9]. PHM during COVID-19 was defined as actions taken by individuals or various groups of people, such as national and local governments, academic and research institutions, communities, and international entities, to slow or stop the spread of infectious diseases [10]. They can exist as non-pharmaceutical groups and are a combination of behavioral, environmental, social, and systems interventions that are used to contain the infection and mitigate disease risk [11,12]. PHM are not limited to public messaging and education but include restrictions to activities and limited access to facilities and institutions, including systems-level interventions (e.g., local and national lockdowns, contact management), and behavioral interventions (e.g., physical distancing/isolation/quarantine, hand hygiene precautions, mandatory use of face masks).

Focusing on patient and public involvement (PPI) can enable the right questions and depth of understanding to exist for problems from the user’s perspective. PPI enhances the efficiency, design, and quality of healthcare initiatives and facilitates decision-making regarding resource allocations and the usability of services by including information about the capabilities, needs, and priorities of local people [13]. PPI has an impact on equity if it succeeds in bringing together diverse communities and social groups and provides a ‘voice’ for marginalized groups, thus reducing the over-representation of some interests. Attention should be directed to the implementation of PPI, as it can have the adverse effect of enlarging the gap between communities and creating greater inequities, particularly among marginalized populations. PPI implementation should accommodate populations who may have limited access to resources and are impacted by varying and unforeseen circumstances [14].

With the use of PPI, not only are the patient and public considered stakeholders, but they are also simultaneously the ones receiving care and are directly affected by their involvement.

## 2. Aim, Objectives and Research Questions

For this study, we used the umbrella review as a research methodology which consists of collecting, reviewing, and analyzing systematic reviews and meta-analyses on a specific research topic [15,16]. The primary aim of this umbrella review was to determine and describe the nature and extent of the patient and public involvement (PPI) in COVID-19 health and social care research. A second aim was to identify research gaps in PPI to inform future studies and research funding priorities.

In this review, the objectives were to:Explore the scientific evidence of incorporating PPI.Summarize the current evidence on COVID-19, including PPI, and determine where and when the research was conducted.Identify the members of the public and the types of patients who participated.Identify any gaps in the literature or study designs that should be further explored and emphasized in future public health measures (PHM) and PPI research.

To help us plan for conducting a structured database that could search for relevant systematic reviews, we developed research questions based on the population, phenomena of interest, and context (PICo) framework. The following research questions followed the population, phenomena of interest, and context (PICo) framework [15] in which the population included the general public regardless of age, gender, and race/ethnicity:Have systematic reviews of PHM during the COVID-19 pandemic reported on assessed PPI activities?What were the PPI activities, and who/what populations were likely to be left out?What are the similarities and differences in the effectiveness of PHM between the systematic review studies?

## 3. Research Design

The findings of reviews pertinent to a research question can be compared and contrasted via an umbrella review [15]. Therefore, this review was conducted using the JBI manual for evidence synthesis for umbrella reviews [16], along with the PRISMA-ScR checklist [17]. Before undertaking this umbrella review, a protocol was registered with PROSPERO: CRD42022307608 [18].

### 3.1. Search Stratgey

Using the PICo framework [19], synonyms, keywords, and controlled vocabulary terms were identified by a public health librarian (N.T.) with expertise in conducting systematic reviews, and this was reviewed for accuracy by the lead author (N.F.) of the research team. These terms were developed under the following concepts: (P) patient and public involvement; (I) public health measures; (Co) COVID-19 and full search strategies are presented in Appendix A. A systematic review filter was adapted from the National Library of Medicine [20] to filter the results by article type. A three-step search approach was used to identify the relevant literature reviews:We conducted preliminary searches to find published reviews in the literature on patient and public involvement (PPI) and public health measures (PHM) during the pandemic. Among the databases searched were the 3iE Database, BMC Systematic Reviews, Campbell Collaboration, Centre for Reviews and Dissemination, Cochrane Library, JBI Evidence Synthesis, PROSPERO, and Google Scholar.In the next stage of the search, nine databases (EBSCO interface—Academic Search Ultimate, APA PsycINFO, CINAHL, Family & Society Studies Worldwide, Health Source: Nursing/Academic Edition, MEDLINE; Epistemonikos, ScienceDirect and WHO COVID-19 Global Literature) were searched on 21 January 2022.Finally, a Google Scholar search was again conducted with the included studies identified from the title/abstract stage. Using the cited by feature, reference lists were checked to identify additional studies that could be appropriate for inclusion.

### 3.2. Inclusion and Exclusion Criteria

The umbrella review included the literature reviews published in English in peer-reviewed journals. Since the pandemic began at the beginning of 2020, the studies were limited to 2020–2022. Any type of literature review was considered (e.g., systematic review, rapid review, integrative review, scoping review, narrative review, meta-analyses, etc.) with studies of any design:Qualitative studies include designs such as grounded theory, ethnography, phenomenology, action research, and qualitative descriptive.Quantitative studies include both experimental (e.g., randomized trials, non-randomized trials) and observational (e.g., cohort, cross-sectional) study designs.We also considered case series, individual case reports, health intervention, or service development.

Studies with human participants were included regardless of age, gender, and race/ethnicity. The interventions/phenomena of interest were public health measures (PHM), and the World Health Organization’s Taxonomy and Glossary of Public Health and Social Measures [10] was used to identify related terms as inclusion criteria for PHM. Studies, including PPI, defined as research carried out by patients and members of the public contributing to the design, implementation, and dissemination of research, were included. Regardless of the geographical location, all settings of the study were included (e.g., healthcare facilities of any type including, but not restricted to schools, in-patients, hospitals, medical centers, community-based care, and long-term care facilities). Non-systematic reviews and individual research studies were excluded.

### 3.3. Study Screening and Selection

Ten graduate students (D.A., L.B., S.L., C.L., E.M., A.M., A.N., L.R., M.D.S., and A.D.S.) organized into five groups independently screened titles/abstracts and then full texts for inclusion. All students involved in reviewing the studies were Master of Public Health and Master of Health Administration graduate students. Several meetings were held prior to reviewing the articles to discuss definitions of PPI and analysis strategies for the purpose of training. Prior to the screening, reviewers were provided with a list of inclusion criteria. This was undertaken to ensure that reviews were conducted consistently among the ten reviewers as a method to minimize reviewer bias. To minimize selection bias, the screening was independently completed by two group members. Discrepancies during the title/abstract and full-text review stages were resolved through discussion among the reviewers within each group. If a consensus could not be reached, a third reviewer (N.F.) was consulted. Throughout the reviews, multiple meetings were conducted to discuss all the decisions completed independently or through a second reviewer. Reviewer and selection bias is possible; however, protocols were in place prior to starting the review process to ensure that these biases were minimal.

### 3.4. Assessment of Methodological Quality/Critical Appraisal of the Included Systematic Reviews

The public health librarian (N.T.) assigned full-text articles identified from the title/abstract review to each group member. These articles were then independently assessed by the other group members for methodological quality using the JBI Critical Appraisal Checklist for systematic reviews and research syntheses [16]. This checklist is designed to check the quality of the systematic reviews, not the quality of the included primary studies. The final appraisal column was calculated for risk of bias by classifications of “N/A”, “No”, and “Unclear”. These were then compared to the risk of bias number ranges—Low (0–3), Medium (4–6), High (7–11)—and assigned to the corresponding level. This umbrella review included all studies, regardless of their methodological quality.

### 3.5. Data Extraction

Data extraction was completed by one reviewer using the JBI Data Extraction Form for the Review of Systematic Reviews and Research Syntheses [16]. Data were independently checked and verified by a second reviewer. A third reviewer (N.F.) was consulted when there was a lack of consensus.

### 3.6. Data Summary

To discuss the findings, the research team compared and contrasted the data extracted from the included studies. An analysis of the data was conducted in a tabular format and was complemented by a narrative synthesis.

## 4. Results

### 4.1. Study Inclusion

The search and screening process can be seen in Figure 1. We obtained a total of 1437 records across all databases and imported these into Zotero: a citation management software [21]. After removing 706 duplicates, a total of 731 distinct records were screened based on the title/abstract in Rayyan: a tool that allows for the independent screening and coding of studies [22]. Fifty-four articles were identified for a full-text review, with a final selection of six systematic reviews that met the inclusion criteria [23,24,25,26,27,28]. In Appendix A, the excluded full-text articles are listed along with their reasons for exclusion (e.g., no patient and public involvement (PPI) or public health measures (PHM) being discussed, or the articles not being review studies).

### 4.2. Methodological Quality of Included Systematic Reviews

We assessed the methodological quality of the six included reviews from the title/abstract stage and reported the findings in Appendix A. Two reviewers appraised the quality of each article by following the JBI Critical Appraisal Checklist for Systematic Reviews and Research Syntheses. Scores were assigned for adherence to each question. The final appraisal column was calculated for risk of bias by classifications of “N/A”, “No”, and “Unclear”. These then were compared to the risk of bias number ranges—Low (0–3), Medium (4–6), High (7–11)—and were assigned to the corresponding level. This umbrella review included six studies, regardless of their methodological quality. After the initial evaluations of articles, disagreements were assessed and resolved by a designated third reviewer. The “Risk of Bias” column is a result of the total number of points scored for each appraisal question. The six reviews were not eliminated according to their results in this tool.

### 4.3. Characteristics of Included Studies

The included six reviews were published in 2021, and their summaries are presented in Appendix A. Our research questions were clearly stated in all the included reviews either in the title, abstract, or in text. A publication bias was not addressed in all the included reviews. Three reviews were determined to have a medium-level risk of bias [23,24,25], and the remaining three reviews were considered to have a low-level risk of bias [26,27,28]. The sources of the included studies and inclusion criteria were appropriate for all reviews. The criteria for appraisal included an appropriate and independent appraisal and was conducted in most of the included studies. The methods used to minimize error in data extraction were not specified in most of the included studies. There were two or more databases searched in all the included studies.

### 4.4. Findings of the Review

Table 1 and Table 2 show the patient and public involvement (PPI) measures and research findings for each of the six studies included in this review. Table 3 provides an overview of the quantitative and qualitative studies and the PPI, and public health measure (PHM) approaches to groups/populations, settings, outcomes, benefits, and gaps identified in the study.

## 5. Summary of Evidence

Table 4 provides a summary of evidence on the effects of patient and public involvement (PPI) in public health measures (PHM) and community involvement included in the studies of our review. The findings and communities involved in each study differed with respect to the behavioral, social, and/or systems interventions that were used to contain the virus and mitigate transmission. Several themes emerged relating to the benefits of PPI, PHM approaches to PPI populations, PPI engagement strategies, and challenges to PPI in health and social research.

### 5.1. Benefits of Patient and Public Involvement (PPI)

During COVID-19 in the United Kingdom, over 4000 local volunteer groups were formed, with as many as three million participants to meet the communities’ needs during the pandemic. In addition, the National Health Service recruited three times their initial target of volunteers, demonstrating that the public wanted to be involved [26]. Mao et al. [26] pointed out that when traditional public services struggled to respond effectively, mutual aid groups played a role in the COVID-19 response. These groups established new partnerships, networks, and knowledge that may continue to be beneficial in the long term. For example, during the pandemic, some of the existing volunteer groups were able to adjust, change focus, and adapt to new needs and challenges.

Adebisi et al. [23] suggested that PPI can work toward improving weak healthcare systems, combating the spread of misinformation, and better meeting the needs of vulnerable populations. Raymond & Ward [28] identified similar findings and expanded further by advising that community engagement was indispensable for designing public health interventions. This allowed for the complexity of individuals with an emphasis on different influences creating inequity and marginalization. Banerjee et al. [24] found PPI to be lacking; however, there was sustained interest on the patient’s and the public’s behalf and a willingness to engage. With improved PPI for the individuals and populations considered at high risk of mortality during the COVID-19 pandemic, there needs to be an improvement in decision making, increased comprehension in regard to the understanding of risks, and increased discussions with healthcare professionals and family members [24].

### 5.2. Public Health Measures (PHM) Approaches

All six review articles discussed the different approaches that communities around the world took in response to mitigating and controlling the COVID-19 pandemic. Ernawati et al. [25] identified an evaluation of society’s awareness around knowledge, attitude, and practice (KAP) in the prevention of COVID-19 as an effective transmission prevention strategy. Similarly, Mao et al. [26] reviewed a combination of behavioral and social approaches and found that community involvement and volunteering were imperative in the public’s response to COVID-19. Adebisi et al. [23] described an approach by South African governments in which they urged citizens to obey COVID-19 precaution measures. Many awareness programs were implemented in order to address pandemic stigmatization, fear, and misinformation related to COVID-19 [23]. They also noted the South African Ministry of Health utilized social media platforms to spread awareness regarding COVID-19. Banerjee et al. [24] found studies that provided publicly available portals, although none were specifically aimed for utilization by the patients and the public. Pegollo et al. [27] discussed the utilization of digital contact tracing (DCT) in response to reducing infection incidence. Lastly, Raymond & Ward [28] noted that local context and community engagement were crucial when implementing public health interventions in order to combat COVID-19.

### 5.3. Patient and Public Involvement (PPI) Population and Engagement Strategies

All community engagement actors and approaches were characterized by a combination of local leaders, community and faith-based organizations, community groups, health facility committees, individuals, and key stakeholders. However, the studies did not include specific equity considerations. Raymond & Ward [28] discussed the impact of COVID-19 on low- and middle-income countries. They identified the barriers created by poverty, inequality, and the inability to comply with lockdown orders. Several adaptive responses were identified in communities throughout different countries, leading the authors to conclude that community engagement holds the potential for future solutions through local context to develop social solutions and effective engagement and communication.

Regarding engagement strategies, Adebisi et al. [23] identified 13 African countries that implemented numerous risk communication and community engagement (RCCE) strategies in response to the decreasing prevalence of COVID-19. Pegollo et al. [27] reviewed digital contact tracing (DCT) in the general population of several countries. The reviewers note that downloading an application was a crucial step in the engagement process and for contract tracing to be successful. Individuals in support of DCT felt it was essential to protect others, and the positive effects outweighed concerns relating to privacy and surveillance. The uptake of DCT was lower in individuals with fewer non-pharmacological interventions, such as mask-wearing and social distancing. Overall, areas of interest regarding engagement in the use of DCT included privacy, government trust/surveillance, cybersecurity, social media sentiment, communication, and usefulness [27].

### 5.4. Challenges to Patient and Public Involvement (PPI) and Public Health Measures (PHM) during the COVID-19 Pandemic

This study included reviews that identified several challenges experienced by communities during the pandemic. Adebisi et al. [23] identified the main challenges to include trust and distrust among the government and officials, cultural, social, and religious resistance, the spreading of misinformation, and continuous issues caused by weak healthcare systems. Raymond & Ward [28] noted challenges relating to misinformation and the fear of being socially denounced, causing pandemic sanctions. An additional challenge was explaining the COVID-19 virus using a combination of different traditional, religious, and scientific ideologies, causing confusion for the public [28]. Pegollo et al. [27] noted the challenge of sharing personal data in healthcare when citizens’ values of privacy had to be overcome. A solution recommended by the authors was to develop a digital solution that was bound by law and respected citizens’ concerns. Mao et al. [26] found many challenges to be centered around constraints and limitations on in-person gatherings and mandating individuals, communities and institutions to adopt unfamiliar practices to continue serving the surrounding communities. Both studies by Mao et al. [26] and Raymond & Ward [28] mentioned the public’s challenge with combating social isolation. Banerjee et al. [24] discovered that the development of COVID-19 mortality risk tools with public engagement was lacking, and there was an absence of mortality risk information designed for patients with underlying conditions. The reviewers found that overall, there was an urgent need to better understand the specific information desired by communities, and further evaluate the impact on vulnerable populations to improve public health. Though traditional modes and models of governance and health systems infrastructures were challenged by COVID-19, regional context and public engagement were essential ideas when creating adequate public health interventions to meet the issues associated with the pandemic [28].

## 6. Discussion

Table 5 shows the overall evidence of patient and public involvement (PPI) in the included reviews. The table is divided into four categories: COVID-19 public health measures, PPI approach and mechanisms, the diversity of PPI stakeholder groups, and the benefits of efficacy that use PPI strategies. Each category is followed by a general combined overview of the studies included in the review.

The purpose of this umbrella review is to provide an overview and synthesis of previously published systematic reviews to determine the nature and extent of PPI in COVID-19 health and social care research to identify where PPI has been used to develop public health measures (PHM). The aim of this umbrella review is to explore scientific evidence for the benefit of incorporating PPI, summarizing the evidence of COVID-19 and PPI to determine where and when the research was conducted, identify members of the public and patients who participated and identify gaps in the literature that need to be further explored.

This umbrella review found that when the public and patients were involved in research practices and processes, valuable perspectives were both generated and found useful for the target audience and community at large. Currently, there is a limited number of studies on PPI during the COVID-19 pandemic, resulting in a gap in the literature relating to this evolving public health topic. The lack of information about the context and process of PPI relating to COVID-19 creates barriers to advancing strategies that can address the pandemic.

While each of the included reviews looked at different measures and levels of patient and public involvement (PPI), there were some commonalities. Through PPI, communities were able to mobilize efforts and identify and adapt to community needs [23,26]. Overall the studies reviewed highlighted PPI involvement in various stages; however, there was a lack of consistent engagement and follow-up throughout the processes [25,26,27]. Furthermore, there were both lacking and inconsistent methods established to report and share effective PPI strategies.

To establish effective community engagement, local organizations need to improve the support provided as well as enhance collaboration and cooperation [23,26]. This may be performed by implementing a follow-up system with the patients and public to ensure their continued involvement, as the decision-making process is critical for everyone involved. Community engagement needs to also better include more geographical settings amongst various populations [28]. Nevertheless, the evidence supports the fact that when barriers were created by poverty and inequality, communities could formulate adaptive responses to COVID-19 [28]. When PPI has an equity-based focus, it allows communities experiencing these inequities to take targeted action, develop solutions, and create conditions for health and well-being. PPI can bring together diverse communities and social groups and provides a ‘voice’ for marginalized populations. As the reviews demonstrated, PPI has the potential to combat the spread of misinformation, resulting in increased trust in public health measures [23,28].

## 7. Limitations

A thorough review was conducted for this umbrella review. However, considering the fact that 1437 records were narrowed down to six articles for inclusion, the relevant studies may have been omitted due to systematic errors during the selection, appraisal, or data extraction process. This umbrella review only includes evidence from the six systematic reviews previously detailed, which may result in desirable details of certain interventions being omitted. In addition, the six articles included in the review addressed PPI in various settings, allowing us to draw conclusions regarding the benefit of PPI as it relates to COVID-19. We relied on the information included in the articles, which is an inherent weakness. For example, an umbrella review is unable to identify a phenomenon that has not been addressed in systematic reviews. An inherent bias may also be a limitation of this review due to one round of appraisal and extraction.

## 8. Conclusions

Figure 2 outlines the existing knowledge on patient and public involvement (PPI) in health and social research, as well as how this paper contributes to it.

In exploring the current and potential ways that we may utilize and benefit from patient and public involvement (PPI) in regard to COVID-19 and public health, we came across many gaps in the literature. A review of the literature presented several opportunities for future practice. Several of the studies indicated that volunteerism, community cohesion, and/or the spontaneous development of assistance programs were prevalent in the public response to COVID-19 [25,26,28]. Policymakers should include these groups when implementing public health measures (PHM). Community and volunteer groups in PPI can be instrumental in gaining public support for PHM and promoting adherence.

Digital technology should be used by policymakers to include the public in data collection and shared decision making. Patient portals or apps can be used to collect public health information and provide feedback. The use of web-based meeting spaces allows for PPI with the inclusion of a wider range of community members and groups when discussing methods of pandemic control and deciding which PHM would be most beneficial to the community. It is critical to respect the privacy concerns of citizens when utilizing digital technology [27].

Some of the included studies also suggest that PHM should be tailored to communities and set within a sociocultural context [23,26,28]. Successful mitigation efforts should be community-specific and have PPI that is integral to shared decision making. This umbrella review highlighted the lack of decision making when patients and the public were involved in PPI, which should be addressed and assessed in further research. Further research should be conducted with PPI to determine which PHM could benefit particular communities and which PHM might cause aversion to adherence. The literature revealed that PHM that did not have widespread community support or did not consider the socio-cultural context were less likely to be followed. Further research is needed to determine the effectiveness of PHM when culturally competent PPI is included in decision making and implementation.

## Figures and Tables

**Figure 1 ijerph-20-04887-f001:**
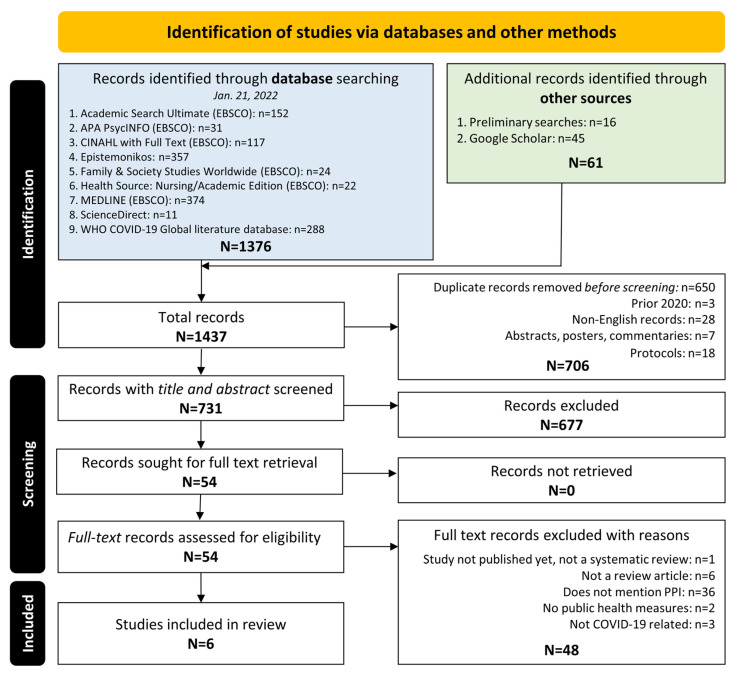
PRISMA flow diagram.

**Figure 2 ijerph-20-04887-f002:**
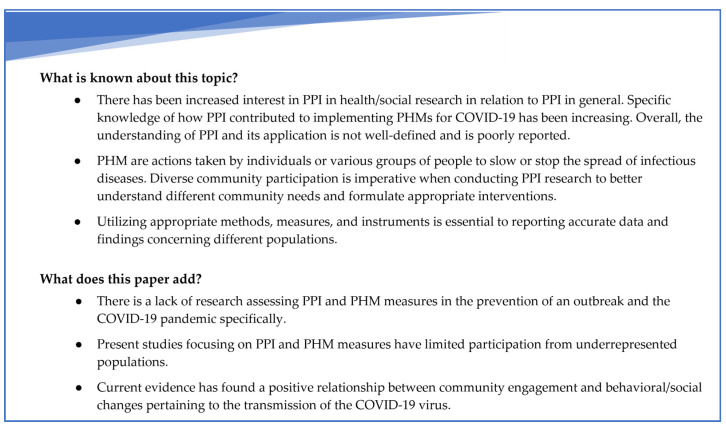
Summary of the existing knowledge on patient and public involvement (PPI) in health and social research and contributions of this umbrella review to it.

**Table 1 ijerph-20-04887-t001:** Quantitative data from included review articles.

Interventions/Phenomena of Interest	Author/Year	Number of Studies/Participants	Results/Findings	Heterogeneity
Risk Communication and Community Engagement (RCCE) Strategies	Adebisi et al., 2021 [23]	Participants from 13 African Countries	PPI measures consisted of in-depth conversations with stakeholders, written feedback, and social media responses. Distrust in the government, weak healthcare systems, widespread rumors and misinformation, the exclusion of some vulnerable groups, and resistance and inertia were challenges to implementing RCCE strategies in African countries. Based on the results, researchers recommend strengthening the strategic mapping of partners, investing in proper coordination structures, resources, training, improving public trust through effective leaderships, ensuring adequate planning, and strengthening the documentation and reporting of activities and experiences for RCCE in Africa.	N/A
COVID-19 Risk Prediction Tools	Banerjee et al., 2021 [24]	53/29.1 million	PPI measures consisted of written feedback from patient and public panels, informal and structured written feedback from stakeholders, user and media feedback on prototype patient portals, and virtual focus groups. Research engagement results have identified a lack of public and patient engagement in COVID-19 risk tools to date and a lack of mortality risk information designs for patients with underlying conditions. Throughout the pandemic, a sustained patient, public interest, and engagement in the development of a risk information tool during and beyond the pandemic was demonstrated. The feasibility to create an online portal containing mortality information with PPI and other healthcare stakeholders was also considered.	N/A
Knowledge, Attitudes, and Behaviors of Communities for the Prevention of COVID-19 Transmission	Ernawati et al., 2021 [25]	10	PPI measures consisted of using community participation to collect data regarding socio-demographic features, knowledge attitudes, and behaviors. Sociodemographic features: older respondents had better attitudes and behaviors towards COVID-19. Women were more likely to perform infection control behavior. Higher-educated respondents had healthier behaviors and unemployed persons had the least healthy behaviors. Knowledge: Increased knowledge leads to increased adherence to public health measures the majority of the time. Increased knowledge led to optimism, limited access to health services, and information affects attitudes/motivation. Attitudes: Self efficacy, view of COVID-19, the government’s idea for handling the pandemic, and social stigma had the most effect on attitudes. Practices: The most utilized practices included washing hands with soap and water, social distancing/self-isolation, and wearing a mask.	High
Volunteering in the UK in the Context of COVID-19	Mao et al., 2021 [26]	27	PPI measures consisted of surveys, interviews, and conversations with volunteers. Volunteer activities consisted of the delivery of essentials, social isolation, support, and supporting improved social determinants of health such as housing. Online volunteering was developed. The expansion of formal volunteering, social action volunteering, and neighborhood support were developed but many models lacked structure and leadership. Middle-aged adults comprised most of the volunteer workforce. A total of 95% of council leaders and chief executives found expanded volunteering to be significant or very significant to their COVID-19 response.	High
Digital Contact Tracing	Pegollo et al., 2021 [27]	41/186,144	PPI consists of user experiences of DCT apps. Accessibility, communication, privacy, cybersecurity, and self-efficacy translated to successful policies.	N/A
Government Mandated Restrictions in Relation to Social and Public Health Adaptive Measures	Raymond & Ward, 2021 [28]	26	PPI consist of intersectoral collaboration, building upon previous epidemic and outbreak experiences, and the establishment of health policy infrastructure, with a focus on community-level engagement. Under-resourced health systems, misinformation, and structural weaknesses contributed to emotional destabilization. Local and contextual strategies including traditional explanatory models and spiritual technologies created resilience.	N/A

**Table 2 ijerph-20-04887-t002:** Qualitative data from included review articles.

Phenomena of Interest/Context	Author/Year	Synthesized Finding	Details of Strategies
Risk Communication and Community Engagement (RCCE) Strategies	Adebisi et al., 2021 [23]	The study aimed at identifying the risk communication and community engagement (RCCE) strategies in 13 African countries. The WHO interim guidance on COVID-19 RCCE, which focused on risk communication systems, coordination among internal personnel and partners, communication and engagement with the public, training, capacity building, and addressing infodemic. Although the African countries utilized support from the WHO. The RCCE response activities were not without challenges. Many of the challenges that were experienced included distrust in the government due to corruption practices, weak healthcare systems caused by limited resources, widespread rumors and misinformation, excluding the specific needs of vulnerable groups, as well as cultural, social, and religious resistance, and inertia.	PPI Measures: engagement with risk communication strategies such as creating operational teams and working groups, engaging with all levels of government, creating risk communication plans, and RCCE budgets. Identifying and establishing relationships with partner organizations and national health authorities to create RCCE coordination and operating strategies. Utilizing community engagement through community leaders and media to understand opinions and beliefs regarding RCCE. Communicating through traditional and non-traditional methods. Training the public on RCCE strategies. Using IT to address infodemic concerns. Face-to-Face: community health organizers, youth-led and women-led movements, home visits, engaging the community, and religious leaders. Media: posters, billboards, radio and television ads, podcasts, social media, WhatsApp Government: Daily press briefings, translation into local languages, contract tracing, hotlines, official websites
COVID-19 Risk Prediction Tools	Banerjee et al., 2021 [24]	The study analyzed mortality risk information for people with ‘high-risk’ conditions for COVID-19. Three themes were identified in the pre- and post prototype release. The three themes are as follows: the information needs of patients, the usability of the information, and shared decision making with healthcare professionals. The findings suggest that there is a lack of public and patient engagement when it comes to COVID-19 risk tools as well as a lack of mortality risk information that is designed for patients with underlying conditions. Another finding showed that there had been sustained patient and public interest and engagement when developing such information during and post-pandemic. The last finding demonstrated the feasibility and utility of one online portal used for mortality information, covering a wide range of conditions, which was informed by patients and the public, ultimately giving context for decision-making and allowing for discussions between health professionals, family members, and caretakers. The findings suggest that multiple stakeholders, methods, and ongoing patient and public involvement and engagement (PPIE) are prerequisites, perhaps requiring a dedicated organization, (such as the Institute for Health Metrics and Evaluation for the Global Burden of Disease Study). Doctors are required to provide ‘all material risks’ when consenting patients, underscoring the need to find new ways to generate and communicate risk information. Second, there is a role for charities, patient organizations, and patients to collaborate and articulate a framework for better risk information across disease silos.	PPI Measures: users assisted in the development and publication of the prototype portal as well as providing feedback for further improvements once the prototype patient portal was published. Informing prototype (March–May 2020) Information Needs: Data were not collected from the tool due to patient concerns about data use and privacy. Usability for Patients: Initial reservations around usability, requiring a further need for dialogue in order to make the tool more useful to patients and the public. Shared decision making: Interest in using mortality risk data in discussions with health professionals was expressed by patients, but there were concerns about how the information would link to COVID-19 advice, and the potential for unintended consequences where the risk could be low. Informing the subsequent development of the public-facing version (May 2020–November 2020) Information needs: Data Representation: specific and simple information related to conditions resulting in multimorbidity. Usability for Patients: a database created for researchers and policymakers that the public has access to with a plan for more patient and public involvement before a user portal is released. Shared Decision-Making: Patients are provided with accurate risk information that is developed in conjunction with specialists and charities, using the data to make informed decisions with their doctors.
Knowledge, Attitudes and Behaviors, of Communities for the Prevention of COVID-19 Transmission	Ernawati et al., 2021 [25]	Knowledge, attitude, and practice factors (KAP) were evaluated in community participation for the prevention of COVID-19 transmission. Knowledge of the COVID-19 pandemic, specifically on modes of transmission, common symptoms, and the prevention of transmission, was found to encourage the public’s participation in preventing transmission. Preventive behavior correlates with the knowledge and attitudes of each community, showing a positive relationship among the three aspects of KAP. Additionally, the results of a good KAP aspect can provide a direct output and, namely, a reduction in the incidence of COVID-19 in each country in a specific time and in the long term.	PPI Measures: data were collected from respondents in individual communities from each country. Findings and conclusions were generalized but also tabulated for specific communities. Knowledge affecting community participation in the prevention of COVID-19 transmission: initial introductory information about COVID-19, information about transmission, how to identify general symptoms, and how to prevent transmission. Attitudes affecting community participation in the prevention of COVID-19 transmission: attitudes describe the government’s view on handling COVID-19, along with individual views which include self-isolation, mask usage, and social distancing. Practices affecting community participation in the prevention of COVID-19 transmission: washing hands properly and appropriately, maintaining physical distance, social distancing, avoiding crowded places, mask usage, and self-isolation.
Volunteering in the UK in the Context of COVID-19	Mao et al., 2021 [26]	Overall, the review suggests that there were diverse models of organization and coordination in COVID-19 volunteering and that community support groups adjusted their activities and scope of action to the perceived needs and challenges. Social networks and connections, local knowledge, and social trust were key dimensions associated with community organizing and volunteering.	PPI Measures: Volunteer profiles, models of a vertical or horizontal organization, volunteer focus, and perceptions and realities of challenges and successes were indicated through the use of qualitative research measures such as surveys, interviews, and conversations. Volunteering Activities: the delivery of essentials such as food and prescriptions, activities to provide support during social isolation, employment assistance, social benefits, mental health, domestic abuse, homelessness, and housing and eviction assistance. Ways to Connect: online activities, Facebook social clubs, youth-led web groups, zoom meetings, WhatsApp, Skype, google docs, and offline activities such as handing out leaflets. Volunteering Models: formal volunteering through already established organizations, social action volunteering through fundraising and donation campaigns, and neighborhood support or grassroot movements. Volunteers: a shift from the traditional elderly population to middle aged adults, primarily women, volunteers who had lower SES, attitudes that included compassion, and trust in others and the government Sustainability: established organizations had lengthy processes to certify new volunteers and often the excitement to volunteer would dissipate before the organizations could process applications, grassroot movements lacked leadership, successful models included giving volunteers the option to decline assignments, giving social recognition, trusting relationships with volunteers, access to funds, and community-led groups that had a positive relationship with the local government with a mutual give-and-take style oversight.
Digital Contact Tracing	Pegollo et al., 2021 [27]	Digital health technologies have the capacity to bring healthcare services to everyone, helping those more vulnerable to feel safe and meaningfully helped while also contributing to public health.	PPI Measures: user interactions, experiences, attitudes, and beliefs were measured by observing and communicating with users. Practicalities: knowledge of the app did not correlate with downloads, a willingness to download the app varied widely across countries, a lack of accessibility, and disproportionately affected vulnerable groups mainly due to SES, education, and the digital divide. Adoption: downloads ranged from 37.3% to 87% and refused or missed downloads ranged from 27.7% to 94.8% across countries, adherence and continued use of the app declined over time, and continued use of the app was more likely to occur amongst people who felt self-efficacy or if their use of the app kept them or their loved ones safe. Participants were more likely to adopt the app with a higher SES, if they lived in affluent neighborhoods, were more educated, had trust in authority, more internet use, more use of pandemic safety measures and coping skills, and if they were men. Effectiveness and Trust: privacy was often considered to be more important than the common social good; a lack of trust in the government prevented effectiveness, concerns about the rapid development of the app and cybersecurity, social media impacted the trust of users whether informed or not, poor communication from the government about the app and its purpose caused barriers to usage whereas governments who educated citizenry showed positive effects.
Government Mandated Restrictions in Relation to Social and Public Health Adaptive Measures	Raymond & Ward 2021 [28]	Innovations and adaptations, through the syntheses of traditional and biomedical discourses and practice, illustrated community resilience and provided models for successful engagement to improve public health outcomes.	PPI Measures: interacting with local leaders, developing a COVID-19 task force, community engagement in contact tracing, “social surveillance”, and volunteerism. Community Cohesion: the management and disbursement of funds/resources became a communal affair, promoting trust, and transparency, and providing much-needed economic relief to families. Adaptive Leadership: A patron-client theory was used to approach the way that local leaders were regarded during the pandemic. The researchers found that the village heads shaped public opinion and that perceptions of COVID-19 served as a consolidating center for volunteers and a conduit for information, and facilitated social assistance.

**Table 3 ijerph-20-04887-t003:** Overview of quantitative and qualitative findings.

Author/Year	PPI Definition	Type of PHM	Approaches Used	Groups Included	Settings	Outcomes	Benefits/Gaps
Adebisi et al., 2021 [23]	Various forms of verbal and written communication	Risk communication and community engagement (RCCE) strategies, such as training and capacity building, risk communication systems, internal and partners’ coordination, community engagement, public communication, contending with uncertainties, addressing misperceptions, and managing misinformation.	The key categories of COVID-19 RCCE based on WHO interim guidance were risk communication systems, internal and partner coordination, community engagement, public communication, addressing infodemic, and training and capacity building.	Community influencers, community health workers and religious leaders, community spokespersons, health professionals, and community health workers.	Ethiopia, Ghana, Kenya, Algeria, Angola, Cote d’Ivoire, the Democratic Republic of the Congo, Mauritius, Nigeria, South Africa, Tanzania, Uganda, and Zambia.	RCCE Strategies to address COVID-19 pandemic in 13 African countries.	Benefits: Given the common RCCE approaches and interventions seen across the continent, it is clear that countries are learning from each other and from global health organizations to develop RCCE programs for COVID-19. Gaps: Challenges with response activities included distrust in the government, cultural, social, and religious resistance, and inertia, as well as widespread fake news and rumors, the exclusion of vulnerable populations, and longstanding issues of weak healthcare systems.
Banerjee et al., 2021 [24]	Including patients and the public in all phases of research development and implementation	Developing a mortality risk calculator, informed by patients and the public, for 87 underlying conditions in the COVID-19 context	Systematic review of published risk tools for the prognosis, provision, and patient testing of new mortality risk estimates for people with high-risk conditions, iterative PPI, and engagement with qualitative analysis.	Patients older than 30 years of age registered with a general practice between 1-1-97 and 1-1-17 with ≥1 year of follow-up data.	Population-based primary care electronic health records.	The study showed a lack of public and patient engagement in COVID-19 risk tools to date and a lack of mortality risk information designed for patients with underlying conditions.	Benefits: Throughout the pandemic, the study demonstrated sustained PPI interest and engagement in the development of risk information tools. The study showed the feasibility and utility of a single online portal for the mortality information of a wide range of conditions, informed by patients and the public. Gaps: Despite research engagement, the results identified a lack of public and patient engagement in COVID-19 risk tools to date and a lack of mortality risk information design for patients with underlying conditions.
Ernawati et al., 2021 [25]	Community participation and willingness to help with disease management activities in respective regions.	Community participation to raise awareness of social distancing and self-isolation.	Systematic review and meta-analysis to determine community knowledge, attitudes, and behavior in preventing the transmission of COVID-19.	Members of the community with varying demographics.	Global review (all regions)	The results of a good knowledge, attitude, and practice aspect can provide a direct output, namely a reduction in the incidence of COVID-19 in each country over a specific time and in the long term.	Benefits: Community knowledge determines how people will behave as it relates to the pandemic.
Mao et al., 2021 [26]	COVID-19 volunteering including both informal and formal volunteering.	Volunteering and community support to aid support self-isolation.	A rapid review of the literature to assess the impact of volunteering at national and local community levels.	Groups including community volunteers.	United Kingdom	The review showed diverse models of organization and coordination in COVID-19 volunteering and that community support groups adjusted their activities and scope of action to perceived needs and challenges. Social networks and connections, local knowledge, and social trust were key dimensions associated with community organizing and volunteering.	Benefits: Community support groups seem to adjust their activities and scope of action to the current needs and challenges. Gaps: Despite the efforts of a few official public institutions and councils, there has been limited community engagement and collaboration with volunteering groups and other community-based organizations.
Pegollo et al., 2021 [27]	Population acceptance and participation in the use of digital contact tracing	Uptake, usage, interaction, and general sentiment or perception of digital contact tracing.	Systematic review of studies reporting on DCT acceptance and indicators, such as knowledge of technology, a willingness to download the DCT app, and the accessibility of the technology.	No age or region-specific limitations were in place.	Global review (all regions).	Adherence and continued use of DCT app declined over time, except among certain groups (individuals who felt self-efficacy, those that felt app usage kept loved ones safe, and those with higher SES).	Benefits: Digital health technologies have the capacity to bring healthcare services to all, helping vulnerable populations feel safe and overall contributing to public health. Gaps: Late adopters are individuals that need the most protection but often lack the equipment and understanding of technology.
Raymond & Ward 2021 [28]	Interacting with local leaders, the development of a COVID-19 task force, community engagement in contact tracing, “social surveillance”, and volunteerism.	Government mandated restrictions on movements to reduce transmission rates in lower/middle income nations.	Systematic review to evaluate the context and construction of community responses, social and psychological effects, the impacts of social and mobility restrictions, health system challenges, and adaptive responses.	No age nor region-specific limitations were in place.	Low- and middle-income countries of the Global South.	Communities worldwide reacted in multiple and complex ways and were influenced by social ruptures, restrictions in social and physical mobility, and ever-looming uncertainties of infection, financial insecurity, stigma, and loss, communities worldwide reacted in multiple and complex ways.	Benefits: Innovations and adaptations, through the syntheses of traditional and biomedical discourses and practice, leading to community resilience and providing models for successful engagement to improve public health outcomes. Gaps: Widespread misinformation and fear of social renunciations resulted in noncooperation with pandemic regulations, aversions, and heightened isolation, allowing the spread of the virus.

**Table 4 ijerph-20-04887-t004:** Summary of evidence for the effect of patient and public involvement (PPI) on public health measures (PHM) and community involvement.

Author/Year	Interventions/Phenomena of Interest	Types of Studies Included in the Synthesis	Synthesized Findings
Adebisi et al., 2021 [23]	Risk Communication and Community Engagement (RCCE) strategies	Peer reviewed articles, reports, newsletters, government documents.	The majority of African countries have implemented risk communication and community engagement (RCCE) strategies to decrease the prevalence of COVID-19.
Banerjee et al., 2021 [24]	COVID-19 risk prediction tools	Living systematic reviews	Even though the public has shown interest in the development of risk information tools, there has been a lack of public and patient involvement in the development of risk prediction tools. There is an urgent need to better understand the specific risk information that patients and the public want overall.
Ernawati et al., 2021 [25]	Knowledge, attitudes, and behaviors, of communities in the prevention of COVID-19 transmission.	Quantitative research, primary data, open access articles, peer-reviewed work.	Evaluation of society awareness around the knowledge, attitudes, practice (KAP) for the prevention of COVID-19 has been found to be an effective transmission prevention strategy.
Mao et al., 2021 [26]	Volunteering in the UK in the context of COVID-19.	Published peer-reviewed articles, reports, briefings, blog posts, newspaper articles, and online media.	Community engagement and adaptation to change during volunteering efforts were essential in the public’s response to COVID-19 in the UK.
Pegollo et al., 2021 [27]	Digital contact tracing	Cross-sectional, population-based controlled experiment, surveys, interviews, text analysis, readability, experiments, longitudinal, comparative mixed methods, app review analysis, app usability, hybrid, and prospective.	Digital health technologies may bring healthcare services to a large population, which can help individuals feel safe while helping contribute to public health. The acceptance of digital contact tracing (DCT) is mainly centered around knowledge, willingness/adherence, usefulness, accessibility, community empowerment, and the concerns of privacy.
Raymond & Ward 2021 [28]	Government mandated restrictions in relation to social and public health adaptive measures.	Empirical, qualitative, field-based, and/or participatory research	Misinformation and fear of being socially chastised resulted in pandemic sanctions, resistance, higher rates of isolation, and increased prevalence rates of the virus. Synthesizing traditional and scientific/medical discourse and practices allowed for innovations and adaptations to communities’ reactions to the pandemic, resulting in community strength and providing methods for successful interactions to improve public health outcomes.

**Table 5 ijerph-20-04887-t005:** Overview of evidence for patient and public involvement (PPI) included in the reviews.

COVID-19 Public Health Measures	PPI Approach and Mechanisms	Diversity of PPI Stakeholder Groups	Benefits of Efficacy of Using PPI Strategies
Evaluation of society awareness around Knowledge, attitude, and practice (KAP) factors. Community involvement/volunteering. Awareness programs aimed at stigmatization, fear, and misinformation. Publicly available portals Digital Contact Tracing (DCT) Local context/community engagement. Social media platforms.	Local context to develop social solutions and effective engagement and communication. Implementation of RCCE strategies. General public using DCT. Mutual aid groups play a role when traditional public service struggle in the COVID-19 response.	Local leaders Community organizations Faith-based organizations Health facility committees Individuals Mutual aid groups	Establishing new partnerships, networks, and knowledge. Being able to adapt to new needs and challenges. Improving weak health care systems. Combating the spread of misinformation. Meet the needs of vulnerable populations. Good for designing public health interventions. Increased comprehension for understanding transmission risk.

## Data Availability

Not applicable.

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
