# Peer review of "COVID-19 Public Health Measures and Patient and Public Involvement in Health and Social Care Research: An Umbrella Review"

_ijerph, 2023, doi:10.3390/ijerph20064887_

Round 1
Reviewer 1 Report
See attached file with my comments

Author Response
Reviewer 1
1- The importance of including stakeholders directly impacted by said issue in research is well documented and should be cited to support this opening statement.
We included the reference.
2- What were those challenges? Was it just time? Was it access because people were ill?
Thank you for the suggestions. We have added the following statement to clarify: “However, the rapid response to the pandemic resulted in PPI being viewed as non-essential leading to minimal inclusion in research, thereby, minimizing the contribution of patients and the public and particularly minority groups in helping find solutions to the pandemic crisis [1]. “
3- I’m concerned about potential plagiarism with this sentence. Is there a citation for this definition?
We paraphrased the quote to “The concept of patient and public involvement refers to conducting research 'with' or 'by' the public, rather than 'to', 'about' or 'for' them. Patients, potential patients, caregivers and people who use health and social care services, as well as representatives of organizations representing such people, are considered the public [2].”
4- What is an example of how poor PPI implementation (I am assuming you mean engagement) leads to greater inequities? Is it a poor sampling strategy?
We added a statement to clarify the implementation approach: ”PPI has an impact on equity if it succeeds in bringing together diverse communities and social groups and provides a ‘voice’ for marginalized groups, thus reducing over-representation of some interests. Attention should be directed to the implementation of PPI, as it can have the adverse effect of enlarging the gap between communities and creating greater inequities, particularly among marginalized populations. PPI implementation should accommodate populations who may have limited access to resources and are impacted by varying and unforeseen circumstances” [14].
5-I think I understand the term umbrella review, but perhaps it would be worthwhile differentiating it from a meta analysis?
We added a statement to clarify “For this study we used the umbrella review as a research methodology which consists of collecting, reviewing and analyzing systematic reviews and meta-analyses on a specific research topic [16, 17].”
6- What do you mean by “map”? Like GIS, or alignment to something?
We deleted ’map’ as we did not include any a map or graphical representation on our findings.
7- This just seems to come out of the blue. Why are you using this framework? Perhaps a line or two explaining what it is might be useful?
We added the below statement to clarify:
“To help us plan for conducting a structured database searching for relevant systematic reviews, we developed research questions based on the Population, Phenomena of Interest, and Context (PICo) Framework. “
8- The multiple use of the term review in one sentence is confusing
To avoid repetition of the word "review", we replaced "review question" with "research question."
9- I’m not sure whether this rewrite now distorts what you are trying to convey. This was my attempt to eliminate the multiple references to reviews in one sentence and to write in the active voice.
To address this concern, we revised the manuscript and changed the passive voice with active voice. Therefore, we kept our original writing within the bulleted list here. The suggested edits in section 3.1 changed the meaning of this sentence. Therefore, we prefer to keep the original language and changed the passive voice to active voice. For example, we restated step 1 as “ We conducted preliminary searches to find published literature reviews on PPI and PHM during the pandemic.”
10- I looked that file over. Will that be included if published? No one will read that document.
According to the JBI Manual for Evidence Synthesis, excluded studies from the full text review must be reported in the supplemental materials.
Aromataris, E.; Fernandez, R.; Godfrey, C.; Holly, C.; Khalil, H.; Tungpunkom, P. Chapter 10: Umbrella Reviews. In JBI Manual for Evidence Synthesis; Aromataris, E., Munn, Z., Eds.; JBI, 2020.
See Appendix 3: https://jbi-global-wiki.refined.site/space/MANUAL/4689980/3.4.12+Review+Appendices
11- Does this section belong in the introduction explaining the benefits of PPI? This doesn’t seem to be pandemic specific?
This statement is a result of one of the six studies included in the review related to the pandemic. We added the below statement to clarify.
With improved PPI for the individuals and populations considered at high risk of mortality during the COVID-19 pandemic, there will be improvement in decision making, increased comprehension in regards to understanding risk, and increased discussions with healthcare professionals and family members [24].
12- Is this now an acceptable term? line 352
Replaced "fake news" with "misinformation"
13- Sections 5.2, 5.3, and 5.4 seem to be the most meaningful. Is there a way to shorten this paper by not having to provide as much detail on how you got to the 6 articles? The crux of the paper seems to be these sections. The pages and pages devoted to the methods divert the reader into thinking this is a methodology article. I like the figure explaining how you got down to 6. Keep that for sure. But there must be a way to cut out all the detail and simply refer to the method you used.
Based on the JBI Manual for Evidence Synthesis, we can not shorten or remove sections of the methods section. See 10.3.7 Methods https://jbi-global-wiki.refined.site/space/MANUAL/4687179/10.3.7+Methods
We cannot remove the Discussion section as it is a requirement based on the JBI Manual for Evidence Synthesis. See 10.3.10 Discussion: https://jbi-global-wiki.refined.site/space/MANUAL/4689253/10.3.10+Discussion
14- I’m guessing these are the headings required by the journal. I find the idea of a discussion and conclusion repetitive. For me, it should just be wrapped up in one section.
You are correct. This is a requirement of the journal, as well as the JBI Manual for Evidence Synthesis:
Journal guidelines (scroll down to the "Research Manuscript Sections": https://www.mdpi.com/journal/ijerph/instructions#preparation
See JBI Manual Chapter 10.3.11 Conclusions and recommendations: https://jbi-global-wiki.refined.site/space/MANUAL/4689291/10.3.11+Conclusions+and+recommendations
Reviewer 2 Report
The reviewed manuscript ijerph-2244711 titled “COVID-19 Public Health Measures and Patient and Public Involvement in Health and Social Care Research: An Umbrella Review” presents a comprehensive analysis of Based on the evidence included, it is evident that patient and public involvement in COVID-19-related research is varied. The existing evidence includes written feedback, conversations with stakeholders, and working groups/task forces. An inconsistent evidence base exists in the application and use of patient and public involvement in public health measures. The authors suggested that the successful mitigation efforts must be community specific while making patient and public involvement an integral component of shared decision-making. This review manuscript presents an interesting view in the context of COVID-19 with broader coverage of applications for public health. The topic of this work is appropriate for the International Journal of Environmental Research and Public Health -MDPI. The synthesized finding work is very well written and the discussion about the topic is relevant in the specific area. As an umbrella review, this review does not offers innovative data but summarizes the varied data in the area. The English language and style is adequate, however, I did not find the option to inform it, thus I selected "English language and style are fine/minor spell check required".
I have listed some minor comments which I feel should be addressed
Specific comments:
1-In my point of view, copy and pasting information is not appropriate in an umbrella review. Please, re-wright with your own words. Line 40-44 “Patient and public involvement entail research being carried out ‘with’ or ‘by’ members of the public, rather than ‘to’, ‘about’ or ‘for’ them. The word public can refer to patients, potential patients, carers and people who use health and social care services, people from organizations that represent people who use services as well as members of the public." [2].
2-Perhaps, the abbreviations PPI and PPM should be excluded to be used in the words. It can be better for general readers.
3-Coud you include more on or two references in line 69 to reference the sentence “a high number of systematic reviews”?
4- In the Study selection the authors inform that: “Ten graduate students (D.A., L.B., S.L., C.L., E.M., A.M., A.N., L.R., M.D.S., and A.D.S.) organized in five groups independently screened titles/abstracts, and then full-text for inclusion. The screening was independently completed by one group member and further reviewed by a different group member.” Could you explain better if the analysis of the studies was similar? Are there any training to the graduate students who have a similar background in this field? If the selection of a study was “further reviewed by a” just one “different group member.”, can you have a bias in the selection? On the other hand, the approach to conducting good research by undergraduate students is interesting.
5- The six studies selected in the PRISMA flow diagram deal with a very different focus, is it possible to get an appropriate conclusion ? It can be included in the Limitations section.
Patient and public involvement can have an impact on the equity of public health. Non-scientifically based data were massively disseminated to society, many times even with the support of governments. In my point of view, this fact strongly affects the patient and public involvement. Perhaps, it can be addressed in the Discussion section.
Author Response
Reviewer 2:
Thank you for your comments.
1-In my point of view, copy and pasting information is not appropriate in an umbrella review. Please, re-wright with your own words. Line 40-44. “Patient and public involvement entail research being carried out ‘with’ or ‘by’ members of the public, rather than ‘to’, ‘about’ or ‘for’ them. Public refers to patients, potential patients, carers, people who use health and social care services, representatives of organizations that represent people who use services, and members of the public." [2].
We paraphrased the quote to the following:
The concept of patient and public involvement refers to conducting research 'with' or 'by' the public, rather than 'to', 'about' or 'for' them. Patients, potential patients, caregivers and people who use health and social care services, as well as representatives of organizations representing such people, are considered the public [2].
2-Perhaps, the abbreviations PPI and PPM should be excluded to be used in the words. It can be better for general readers.
We replaced in some instances the acronyms with the full names “patient and public involvement” and “public health measures” whenever appropriate for better readability. For example, we introduced the acronyms at the beginning of a section, then continued with the acronym within this section to allow continuity and easier referencing to the full name within the same section.
3- Could you include more on or two references in line 69 to reference the sentence “a high number of systematic reviews”?
Thank you for this suggestion. However, we cited the study [7] on line 71, where we found the “high number of systematic reviews.” The authors of this study identified 564 records for their analysis thus making it impossible for us to cite these in our paper.
4- In the Study selection the authors inform that: “Ten graduate students (D.A., L.B., S.L., C.L., E.M., A.M., A.N., L.R., M.D.S., and A.D.S.) organized in five groups independently screened titles/abstracts, and then full-text for inclusion. The screening was independently completed by one group member and further reviewed by a different group member.” Could you explain better if the analysis of the studies was similar? Are there any training to the graduate students who have a similar background in this field? If the selection of a study was “further reviewed by a” just one “different group member.”, can you have a bias in the selection? On the other hand, the approach to conducting good research by undergraduate students is interesting.
Thank you for this suggestion. We added more details about training of students and the analysis process to section 3.3. The section has been re-written to the following:
Ten graduate students (D.A., L.B., S.L., C.L., E.M., A.M., A.N., L.R., M.D.S., and A.D.S.) organized in five groups independently screened titles/abstracts, and then full-texts for inclusion. All students involved in reviewing the studies are Master of Public Health and Master of Health Administration graduate students. Several meetings were held prior to reviewing articles to discuss definitions of PPI and analysis strategies, for the purpose of training. Prior to screening, reviewers were provided with a list of inclusion criteria. This was done to ensure that reviews were conducted consistently among the ten reviewers as a method to minimize reviewer bias. To minimize selection bias, the screening was independently completed by two group members. Discrepancies during the title/abstract and full-text review stages were resolved through discussion among the reviewers within each group. If a consensus could not be reached, a third reviewer (N.F.) was consulted. Throughout the reviews, multiple meetings were conducted to discuss all decisions completed independently or through a second reviewer. Reviewer and selection bias are possible, however, protocols were in place prior to starting the review process to ensure biases were minimal.
5- The six studies selected in the PRISMA flow diagram deal with a very different focus, is it possible to get an appropriate conclusion? It can be included in the Limitations section.
We included a statement in the limitations section regarding the studies of PPI in various settings allowing us to draw conclusions regarding benefit.
Patient and public involvement can have an impact on the equity of public health. Non-scientifically based data were massively disseminated to society, many times even with the support of governments. In my point of view, this fact strongly affects the patient and public involvement. Perhaps, it can be addressed in the Discussion section.
Thank you for this suggestion, we added the following statement to discussion to address the impact on equity. “When PPI has an equity-based focus, it allows communities experiencing these inequities to take targeted action, develop solutions and create conditions for health and well-being. PPI can bring together diverse communities and social groups and provides a ‘voice’ for marginalized populations. As the reviews demonstrated, PPI has the potential to combat the spread of misinformation resulting in increased trust in public health measures [23,28]”.